# Current Self-Healing Binders for Energetic Composite Material Applications

**DOI:** 10.3390/molecules28010428

**Published:** 2023-01-03

**Authors:** Jing Yang, Zhehong Lu, Xin Zhou, Zhe Sun, Yubing Hu, Tianfu Zhang, Chao Wu, Guangpu Zhang, Wei Jiang

**Affiliations:** 1National Special Superfine Powder Engineering Technology Research Center, Nanjing University of Science and Technology, Nanjing 210094, China; 2Science and Technology on Aerospace Chemical Power Laboratory, Hubei Institute of Aerospace Chemotechnology, Xiangyang 441003, China; 3School of Material Science and Technology, Nanjing Institute of Technology, Nanjing 211167, China

**Keywords:** energetic composite materials, cracks and defects, self-healing, binders, dynamic chemistry

## Abstract

Energetic composite materials (ECMs) are the basic materials of polymer binder explosives and composite solid propellants, which are mainly composed of explosive crystals and binders. During the manufacturing, storage and use of ECMs, the bonding surface is prone to micro/fine cracks or defects caused by external stimuli such as temperature, humidity and impact, affecting the safety and service of ECMs. Therefore, substantial efforts have been devoted to designing suitable self-healing binders aimed at repairing cracks/defects. This review describes the research progress on self-healing binders for ECMs. The structural designs of these strategies to manipulate macro-molecular and/or supramolecular polymers are discussed in detail, and then the implementation of these strategies on ECMs is discussed. However, the reasonable configuration of robust microstructures and effective dynamic exchange are still challenges. Therefore, the prospects for the development of self-healing binders for ECMs are proposed. These critical insights are emphasized to guide the research on developing novel self-healing binders for ECMs in the future.

## 1. Introduction

Energetic composite materials (ECMs) are the basic materials of polymer binder explosives (PBXs) and composite solid propellants (CSPs), as well as the symbol and key technologies of upgraded weapons. They are a class of high-filled polymer composites, mainly composed of explosive crystals and a small number of polymer binders (5~20 wt%). Under external stimulation (temperature, humidity, vibration, etc.), the bonding surface is prone to form micro/fine cracks, defects and other damages [1,2]. On the one hand, the existence of microcracks is not conducive to the mechanical properties of ECMs. On the other hand, microcracks and defects tend to cause excessive hot spots, which increases the risk of explosion when subjected to external stimuli (such as impact or friction) [3]. Introducing the function of self-healing to ECMs will solve the problems mentioned above, and the stability, security and service life of ECMs will, therefore, be improved. It can be predicted that the research on self-healing ECMs will gain increasing attention in the coming future.

In order to repair hidden microcracks, increase safety and prolong the service life, the concept of self-healing polymers was proposed in the 1980s [4]. Conceptually, self-healing polymers have the inherent ability to substantially recover the load transfer capability after damage. This recovery can occur autonomously or by applying specific stimuli (such as radiation, heat, or water). Therefore, these materials are expected to make a remarkable difference in the improvement of the durability and safety of polymer materials without requiring extra external maintenance or expensive active monitoring [5]. In recent years, a large number of self-healing polymers have been developed [6,7,8].

According to the self-healing strategy, self-healing materials can be divided into external self-healing materials and intrinsic self-healing materials. The external self-healing process mainly depends on microcapsules [9,10,11], microvascular networks [12,13,14] or hollow fibers [15,16,17] that are added to the polymer matrix. Therefore, the external self-healing process requires a high cost of healing agents to allow limited repair times, which restricts the application in self-healing ECM systems. The intrinsic self-healing process is generally based on the dissociation and rearrangement of dynamic covalent or non-covalent bonds in materials, which can be spontaneous or driven by external stimuli (heat, light, pH, etc.). Dynamic covalent bonds include Diels–Alder bonds [18,19,20], disulfide bonds [21,22,23], acyl semicarbazides (ASCZ) [24], acylhydrazone bonds [25,26,27], imine bonds [28,29,30], borate ester bonds [31,32,33], diselenide bonds [34,35,36], etc. Dynamic non-covalent bonds include hydrogen bonds [37,38,39], metal–ligand coordination [40,41,42], host–guest interactions [43,44,45], donor–acceptor interactions [46,47,48], ionic bonds [49,50,51], etc.

In intrinsic self-healing materials, these dynamic covalent and non-covalent bonds have the potential to provide self-healing, which can both be autonomous and experience several healing cycles, inducing the fusion between material science and supramolecular chemistry [52]. Owing to the fact that the relevant interactions of self-healing are dynamically reversible, the repair of the intrinsic self-healing materials can be carried out many times compared to the external self-healing materials. Therefore, researchers tend to select the intrinsic self-healing binders to apply to ECMs. Currently, self-healing methods for ECMs mainly involve several dynamic chemistries (Figure 1). When ECMs are damaged, dynamic covalent or non-covalent bonds in the binders can be reconnected to repair cracks or defects. This review describes the research progress on intrinsic self-healing binders in ECMs systems. The structural design of these strategies to manipulate macromolecular and/or supramolecular polymers will be discussed in detail. Then, the implementation of these strategies on ECMs will be discussed. Finally, the development directions of these self-healing material systems will be proposed. Solving the problems of crack-healing in the field of ECMs will require the cooperation of scientists from different disciplines. All these scientists aim to develop new advanced technologies, which will help improve the stability, safety and service life of ECMs.

## 2. Dynamic Covalent Bonds in Self-Healing ECMs

Covalent bonds possess higher fracture tolerance than non-covalent bonds, which can endow the polymer matrix with stronger mechanical properties. Self-healing polymers based on dynamic covalent bonds can undergo dynamic bond dissociation and rearrangement under external stimuli. Due to the stability of bonds and the high efficiency of the reversible reaction, self-healing polymers can heal the damaged parts independently.

### 2.1. Diels–Alder Reaction

The Diels–Alder reaction is a cycloaddition reaction, that is, the reaction of conjugated dienes with substituted olefins (dienophiles) to form cyclohexene adducts [53]. Common conjugated dienes include furan, furan amine, furan alcohol, furan mercaptan and tetrahydrofuran methacrylate, and frequent dienophiles include maleic anhydride, maleimide and bismaleimide [54,55,56]. The Diels–Alder reaction is thermoreversible, for which the degree of the reaction can be controlled using temperature. Therefore, this mechanism is suitable for self-healing polymers.

Liang et al. [57] designed a TFP (trifuryl propane)-FTPB (furyl terminated polybutadiene)-PDMI (N, N′-1,3-Phthalic maleimide) self-healing binder based on the Diels–Alder reaction. The synthesis methods of TFP and FTPB are shown in Figure 2a. The addition of TFP improved the perfection of the polymer cross-linking network, which is conducive to tensile strength. The existence of Diels–Alder bonds ensured the healing of cracks (Figure 2b). At 120 °C, the occurrence of the Retro-Diels–Alder reaction caused the separation of diene and dienophile, and the molecular mobility increased as the molecular chain became shorter. Active molecular diffusion is beneficial to the rearrangement of molecular chains. At 60 °C, the reassociation of Diels–Alder bonds further repaired the mechanical damages of materials. The self-healing process could be observed directly using the hot-stage optical microscope. As shown in Figure 2c, the crack width was significantly reduced compared with that before healing, which is mainly related to the thermal reversibility of Diels–Alder bonds.

Subsequently, Xia et al. [58], based on Liang’s work, synthesized a novel FTPB using the reaction of isocyanate terminated polybutadiene with fury amine, and then prepared self-healing binder films (FTPB-DAs) based on the Diels–Alder reaction system using the reaction of furan with bismaleimide. The FTPB-DA could be converted to FTPB and bismaleimide at 120 °C. When the temperature dropped to 60 °C, it would be re-crosslinked to form FTPB-DA, thus showing that it has a self-healing property (Figure 3a). In Xia’s strategy, the content of furan groups on the FTPB skeleton could be adjusted by changing the ratio of -NCO and -OH, so as to regulate the chemical crosslinking density of the binders, which is convenient for further balancing the mechanical properties and self-healing properties. Binders with thermal reversibility were further utilized with HMX to prepare PBX (DAPU-HMX) [59]. A CT test showed that, after impact, three cracks formed in the damaged sample, which gradually reduced or even disappeared after healing (Figure 3b). The Brazilian test results indicated that the mechanical properties of the damaged DAPU-HMX samples were seriously reduced, and the healed mechanical strength could recover to 85% of the initial (Figure 3c,d). Thermally reversible Diels–Alder bonds have been shown to be effective for the self-healing of ECMs. The rearrangement of molecular chains and the reassociation of Diels–Alder bonds at the crack repair the damaged network and form new “topological entanglements”. The “topological entanglements” further restore the mechanical properties of the materials. However, for the self-healing binders based on the Diels–Alder reaction, the activation temperature of the Retro-Diels–Alder reaction that is responsible for the repair process is as high as 100~130 °C. For sensitive energetic materials, safety needs further evaluation. Therefore, researchers tend to apply mild dynamic chemistries to ECMs.

### 2.2. Disulfide Exchange Reaction

Disulfide bonds are ubiquitous in organisms and are mainly used to maintain the tertiary structure of proteins, which can be dissociated under the action of light, heat or mechanical force [60,61], or be reformed and exchanged in the appropriate temperature and pH environment (Figure 4a) [62,63]. In addition, disulfide exchange provides significant advantages in self-healing, because S–S bonds can be combined into networks with low glass transition temperature, promoting low-temperature reversibility [64].

CSPs are mainly composed of polymer binders, oxidizers, combustion promoters, aluminum powder and other components [65,66]. In order to ensure high energy, the highly energetic crystals in CSPs usually exceed 80 wt%, while polymer binders are mostly less than 20 wt% [67]. In other words, CSPs belong to a kind of high-filling polymer composite. Such compositional characteristics usually result in low strength and high brittleness. In practical application, the crystal layer is easy to peel off from the coating due to the influence of a complex environment such as external temperature and stress, which leads to the generation of cracks/voids [68,69]. The rearrangement and exchange of disulfide bonds could promote the reattachment of coatings and grain layers, thus healing the debonding interface (Figure 4b) [70]. Furthermore, dynamic disulfide bonds can exhibit high dynamics at lower temperatures, so they are also suitable for repairing cracks/defects inside ECMs. Li et al. [71] prepared a new polyurethane binder (DSPU) with polycaprolactone glycol as the molecular skeleton and bis (4-aminophenyl) disulfide as the chain extender to solve the micro damage problem in PBXs (Figure 4c). Due to the covalent dynamic disulfide bonds in the binders, heat drives in situ healing at the crack/defect. The damaged CL-20-based PBXs just require being heated to 60 °C for several hours to repair, which is obviously related to the dynamic chemical properties of disulfide bonds under heating conditions.

**Figure 4 molecules-28-00428-f004:**
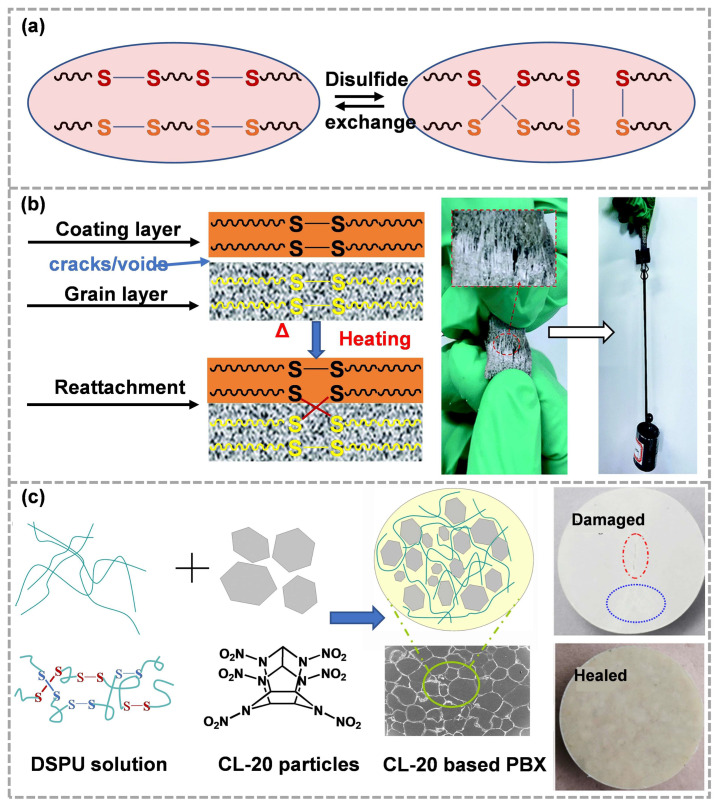
(**a**) Disulfide chain exchange (reproduced from Ref. [63] with permission; Copyright Copyright Springer Nature, 2020), (**b**) the self-healing of the debonding interface between the grain and coating layer(reproduced from Ref. [70] with permission; Copyright Royal Society of Chemistry, 2020), (**c**) the preparation and the damage self-healing of CL-20-based PBX using DSPU as binder(reproduced from Ref. [71] with permission; Copyright Elsevier, 2019).

An energetic binder is a kind of polymer with a large number of energetic groups on molecular chains [72,73,74]. Compared to traditional inert binders, energetic binders possess not only the basic performance of binders, but also the energy properties [75,76]. Among the most energetic groups, -N_3_ has a high positive formation enthalpy (+355.6 kJ mol^−1^). The introduction of -N_3_ into the polymer will not affect the original hydrocarbon ratio. Moreover, -N_3_ could perform thermal decomposition independently before the main chain, which not only increases the energy density of the binder, but also facilitates the decomposition of ECMs to some extent [77]. Glycidyl azide polymer (GAP) is a hydroxy terminated polymer with a large amount of -N_3_, which is usually obtained using modification and azidation of poly(epichlorohydrin) [78,79,80]. In order to further improve the energy level of ECMs, GAP, as an energetic polyether, has gradually attracted attention. Compared with the inert binders, the ECMs with GAP-based binders showed excellent combustion performance (Figure 5a) [81].

Hu et al. [82] utilized GAP, 2-hydroxyethyl disulfide (HEDS) and trimethylolpropane (TMP) as a soft segment, chain extender and cross-linking agent, respectively, to prepare a series of polyurethane vitrimers (GAPUVs). Using an optimization of the crosslinking density and composition of thermosetting GAPUVs, the mechanical properties were significantly improved. With the addition of dynamic disulfide bonds, GAPUVs showed obvious healing ability (Figure 5b) and reprocessing ability after mild heating. Furthermore, the scratch-healing efficiency of the ECMs with GAPUVs (binders) and aluminum powder could exceed 95%. Subsequently, Ding et al. [83] synthesized a self-healing energetic linear polyurethane elastomer (EPU-SS) based on disulfide bonds (Figure 5c). The elastomer was prepared using a two-step method and possessed high self-healing efficiency and mechanical properties, which were attributed to the carefully designed surface energy driving and dynamic hard domains. Then, based on the physical model of interface healing, the variation trend in surface tension, crack bottom radius and depth during the healing process was calculated, and the mechanism of interface healing was obtained. The polyurethane elastomer with low crosslink density could generate excess surface energy at the damage sites to drive the self-healing process, and the addition of a small number of disulfide bonds could further reduce the healing energy barrier. Overall, high filler loading will improve the hardness of polymer composites but will also hinder the process of interface healing. Therefore, the healing ability and mechanical strength of ultra-high-filling polymer composites are contradictory and difficult to optimize at the same time. Ding et al. [84] used EPU-SS as a binder to prepare ECMs with RDX and aluminum powder, which developed a crack-healing method (Figure 5d). The compressive stress at room temperature significantly increased the interface contact effect. Then, the adhesion effectively closed the crack, the surface energy driving promoted the movement of the polymer chains and the reversibility of disulfide bonds helped to rebuild a new polymer network at the interface, thus showing that the cracks could be repaired. In the future, it will be more practical if the reasonable configuration of mechanical properties and efficient self-healing ability can be realized to simplify the crack-healing process.

### 2.3. Dynamic Chemical Reactions of Other Covalent Bonds

Acyl semicarbazide (ASCZ) is a combination of urea and amide linked by an N–N bond, which can be easily formed using the addition reaction of isocyanate and hydrazide [24]. The ASCZ motif is dynamic, which can form an activated n-center transition state, promote proton transfer and, thus, reduce the dissociation energy barrier in ASCZ motifs [85]. Under high temperature, the ASCZ group can reversibly generate isocyanate and hydrazide (Figure 6a) [24], which shows thermal reversibility, providing a new direction for the molecular engineering design of high-performance dynamic polymers. However, similar to Diels–Alder bonds, the rapid dissociation of ASCZ groups needs to be carried out at ~120 °C. The difference is that, in addition to the chemical dynamics, ASCZ groups could provide multiple hydrogen bond donors and acceptors, which will be detailed in Section 3.1.

As a major group element, selenium has similar chemical properties to sulfur. It is worth noting that the bond energy of the Se–Se bond (172 kJ mol^−1^) is lower than that of the S–S bond (240 kJ mol^−1^), which means that the selenium bond, as a dynamic covalent bond, can respond to more mild stimuli, thus stimulating new molecular engineering [86,87]. It has been proven that the dynamic exchange reaction of the selenide bond can take place under heating or visible light irradiation (Figure 6b) [88]. It is reported that the dynamic recombination of the selenide bonds could be effectively used to restore the integrity of the asphalt network at fracture (Figure 6c) [34], providing a new design strategy for ECMs with ultra-high filling. Compared to disulfide bonds, lower bond energy reduces the energy barrier of self-healing. It can be predicted that the dynamic exchange of selenium bonds at the crack/defect can realize the autonomous repair of composite materials. However, low bond energy is not conducive to mechanical properties. Great efforts are still needed to balance network dynamics and robustness.

**Figure 6 molecules-28-00428-f006:**
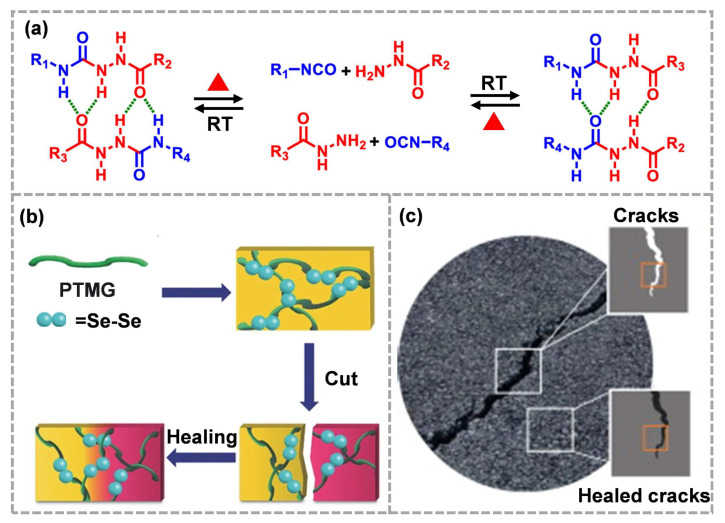
(**a**) Dynamic chemistry of ASCZ groups (reproduced from Ref. [24] with permission; Copyright American Chemical Society, 2020), (**b**) visible-light-induced self-healing process of diselenide-containing polymer (reproduced from Ref. [88] with permission; Copyright Wiley, 2015), (**c**) crack healing of composites using diselenide-containing polymers as binders (reproduced from Ref. [34] with permission; Copyright Elsevier, 2021).

## 3. Dynamic Non-Covalent Bonds in Self-Healing ECMs

Contrary to covalent bonds, reversible non-covalent bonds are characterized by low bond energies. When the polymer network is destroyed by external stimuli, the relatively weak non-covalent bonds will be broken. Then, with the rearrangement/association of dissociated non-covalent bonds, the polymer network can be reconstructed [52,89]. Self-healing polymers based on reversible non-covalent bonds have many advantages, such as reversibility, directivity, sensitivity, etc. Because of the low bond energy, it is a challenging task to achieve excellent mechanical properties and high self-healing efficiency [90,91]. This section mainly introduces several methods that have been applied in ECM systems, including hydrogen bonds, metal–ligand coordination and ion interaction.

### 3.1. Hydrogen Bonds

Up to now, due to the directivity, versatility and reversibility, hydrogen bonding has been one of the most widely used non-covalent interactions for creating physically crosslinked polymer materials with adjustable microstructures and customized properties [92]. Using external stimulation (light and heat), the dissociation/rearrangement of hydrogen bonds can be realized, thus helping to repair the damaged cross-linked network [93,94].

The bond energies of most hydrogen bonds are much lower than those of covalent bonds. For example, a single hydrogen bond formed between carbamates is only 36 kJ mol^−1^, the double hydrogen bonds of urea groups can reach 67 kJ mol^−1^, and for 2-ureido-4[1H]-pyrimidinone (UPy), the bond energy can reach 160 kJ mol^−1^ (Figure 7a) [95,96]. For these reasons, many new strategies have emerged for the design of supramolecular chemistry. The single hydrogen bond is not enough to induce supramolecular self-assembly behavior, but multiple hydrogen bonds formed in an oriented manner can provide a strong assembly binding force. However, although hydrogen bonding units with strong binding energy have good thermodynamic stability, they may inhibit the dynamics of polymer networks and even form crystallization, which is not conducive to self-healing [97]. Fortunately, the inhibition of this crystallization behavior can be easily achieved using molecular engineering. For example, the zigzag arrangement of hydrogen bonds from thiourea does not cause crystallization of the polymer chains [97]. In addition, weak hydrogen bonds from thiourea can cooperate with strong hydrogen bonds from urea groups to build a robust supramolecular dynamic network, where strong hydrogen bonds are responsible for mechanical strength and weak hydrogen bonds are responsible for self-healing [98].

Based on the strategy of hierarchical hydrogen bonds, a self-healing polymer composed of urethane and urea groups can also be synthesized using a one-step method [99]. “The dynamic hard domains” assembled by hierarchical hydrogen bonds not only provide the necessary mechanical strength, but they also provide excellent toughness and room temperature self-healing capability through the continuous breaking of hard domains and rapid rearrangement of networks. When such polymers were used as binders in the preparation of ECMs, long linear polyurethane chains intertwined to form polymer networks, which could bond with energetic crystals. Using the self-repair function of the binders, the internal damage of ECMs could be partially repaired (Figure 7b) [100]. However, in self-healing ECMs, great efforts are still needed to achieve strong mechanical properties and high repair efficiencies.

The self-healing energetic binders, which have both energetic units and self-healing functions, can endow ECMs with the ability to repair cracks and contribute positively to the overall energy level. However, the existence of large-size side groups on the soft chains is not conducive to mechanical properties. To solve this problem, we have developed an energetic self-healing binder based on hierarchical and multi-phase hydrogen bonding, which possessed strong adhesion, good energy properties and rapid room-temperature self-repair capability (Figure 7c) [81]. Enhancing the interface between the adhesive and explosive crystal can make the adhesive “rivet” onto the crystal surface, thus improving the strength and toughness [101]. The excellent room-temperature self-healing ability is due to abundant dynamic hydrogen bonds between the hard and soft phases, and the good energy level is attributed to a large number of -N_3_ on the polymer skeleton. The strong adhesion is mainly inspired by the *Parthenocissus* with multiple suckers. Plentiful hydrogen bond sites on the molecular chains of the soft phase can strengthen the interface between the binders and the explosive crystals, similar to “suckers”, thus ensuring the good mechanical properties of ECMs.

Multiple hydrogen bonding is an effective method for strengthening and toughening, which has stronger interaction than a single hydrogen bond [102]. UPy is a typical multiple hydrogen bonding unit, which can form stable dimers by hydrogen bond interaction. The structure of UPy was found by Meijer et al. [103], which can form strong quadruple hydrogen bonds during dimerization, and the polymerization constant was as high as 6 × 107 M^−1^. The dimer of UPy is thermally reversible. Under external stimulation (such as heating), the hydrogen bonds between the dimers of UPy breaks and the dimer dissociates. At low temperatures, hydrogen bonds rearrange and dimers regenerate [104]. Therefore, the UPy units can be used as non-covalent dynamic bonds to make the materials show good self-repair properties and mechanical properties.

Section 2.3 introduced that ASCZ groups have a thermoreversible dynamic structure [105]; in addition, they can provide multiple hydrogen bond donors and acceptors for hard domains. Unlike UPy, which forms hydrogen-bonded dimers, ASCZ groups may participate in trimers, tetramers or even larger hydrogen-bonded aggregates (Figure 7d) [24,94]. This structure is conducive to the formation of more dense hydrogen bond arrays, which can greatly improve tensile strength and toughness [106]. However, the tightly arranged hard segments will reduce the dynamics of hydrogen bonds, leading to higher self-healing temperatures (>100 °C), which limits the application in ECMs. Therefore, on the premise of meeting the mechanical strength of ECMs, it will be a major research direction in the future to improve the dynamics of polymer networks and endow ECMs with efficient crack-repair capability. We recently reported a new design strategy for a strong, tough, and effective self-healing energetic binder [107]. First, multiple hydrogen bonds from ASCZ motifs enhance the stability of polymer networks. Then, by implanting asymmetric alicyclic structures in the hard segments, the density of the hydrogen bond array is reduced, and the size of hard domains is micro-scale trimmed so that the network dynamics and molecular chain mobility are improved. Finally, ECMs with the designed energetic binders show excellent tensile strength (8.85 MPa) and toughness (12.49 MJ m^−3^). In addition, damaged ECMs can be repaired after being placed at medium temperature (Figure 7e). Additional engineering strategies are still needed to further reduce the energy barrier of dynamic rearrangement while ensuring the mechanical properties.

**Figure 7 molecules-28-00428-f007:**
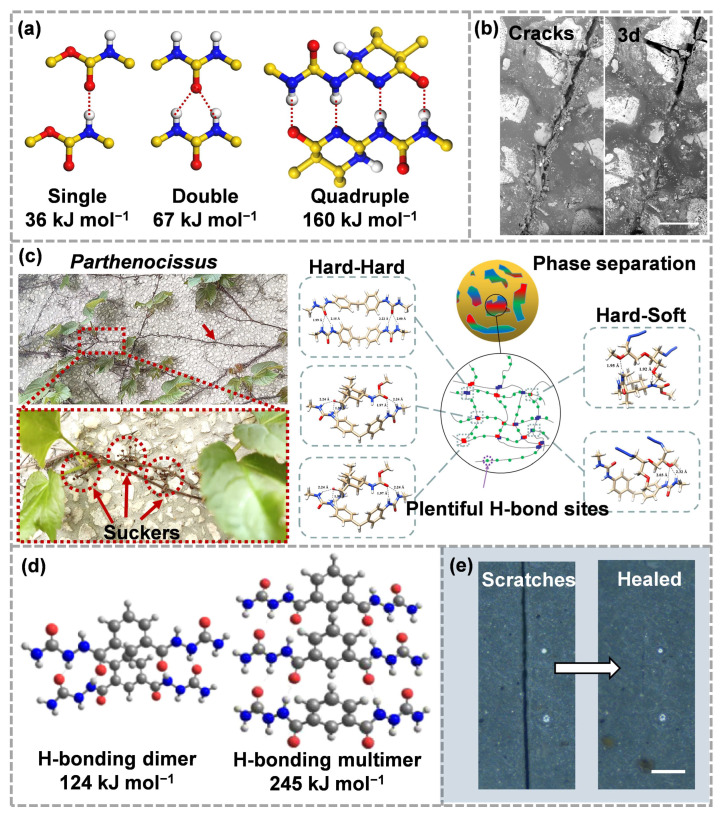
(**a**) Bonding energies of single, double(reproduced from Ref. [96] with permission; Copyright Royal Society of Chemistry, 2022) and quadruple hydrogen bonds (reproduced from Ref. [95] with permission; Copyright American Chemical Society, 2011), (**b**) scratches on the propellant containing hydrogen bonds as binders before and after self-healing(reproduced from Ref. [100] with permission; Copyright Royal Society of Chemistry, 2021), (**c**) schematic illustration of *Parthenocissus* with abundant suckers and chemical structures of a Parthenocissus-like self-healing energetic binder (reproduced from Ref. [81] with permission; Copyright Royal Society of Chemistry, 2021), (**d**) different H-bonded aggregates of ASCZ moieties and their corresponding stabilization energy (reproduced from Ref. [24] with permission; Copyright American Chemical Society, 2020), (**e**) optical microscope images of ECMs containing ASCZ moieties energetic adhesive demonstrating self-healing (reproduced from Ref. [107] with permission; Copyright Elsevier, 2023).

### 3.2. Metal–Ligand Coordination

Metal–ligand coordination is a very unique non-covalent interaction, consisting of metal ions (coordination centers) and surrounding organic molecules (ligands) [108]. In the field of ECMs, metal–ligand coordination can be used to improve the film-forming ability and the adhesion between ECMs and substrates [109]. Generally speaking, the strength of the coordination bond is between the covalent bond and the van der Waals force. Most importantly, the strength of the coordination bond can be adjusted in a large range (about 25~95% of covalent bonds) [110,111]. Sometimes bonds are thermodynamically stable but dynamically unstable [112,113]. Using an optimization of the ratio of ligands to metal ions, the bonding strength can be adjusted to a weak dynamic force, which is conducive to self-repair [114]. In addition, the presence of functional metal ions and/or ligands can stimulate various functions, so it is easy to find valuable potential applications in various technical fields. For example, pyridine–copper coordination has a strong photothermal effect, which can rapidly raise the temperature of the sample to 60 °C under near-infrared light irradiation [115]. At this time, the sticky reptation modes of supramolecular polymer are activated, which drives network reorganization and rapid self-repair. This strong photothermal effect of metal ions and ligands provides a new idea for simplifying self-healing procedures and reducing repair costs.

For most self-healing materials, there is usually a trade-off between mechanical performance and dynamic repair, that is, stronger interactions usually lead to robust cross-linked networks that are less dynamic [116,117]. In general, thermodynamic stability supports the mechanical properties of polymers, and dynamic instability accelerates self-repair. Therefore, it is important to design a cross-linking network with thermodynamic stability and dynamic instability to achieve both high strength and rapid self-healing. Recently, our group reported a kind of energetic binder (3,5-BTP-PDMS-Zn) based on metal-ligand coordination (Figure 8a), which can rapidly repair ECMs at low temperatures [118]. On the one hand, 3,5-bis (1,2,3-triazol-4-yl) pyridine (BPT) could be used as an energetic unit to endow the binder with energy properties. On the other hand, the metal–ligand coordination between Zn^2+^ and BPT could serve as thermodynamically stable and dynamically unstable cross-linking sites. The thermodynamic stability endowed the binder with mechanically robust properties, and the dynamic exchange of the coordination bonds helped to achieve autonomous self-healing (Figure 8b). It is worth noting that relatively short molecular chains (Mn = 7372) and extremely low glass transition temperature (−138.7 °C) ensured the mobility of polymer chains, further promoting the autonomous self-repairing of damaged ECMs at low temperatures (Figure 8c).

### 3.3. Ionic Interaction

An ionic liquid is a kind of liquid salt, whose melting point is generally lower than a specific temperature [119]. Ionic liquids and poly-ionic liquids have attracted extensive attention due to their high conductivity, electrochemical stability, flame retardancy and negligible volatility [120,121]. In particular, the supramolecular ion interaction has high stability and easy accessibility, which promotes the development of self-healing materials [122,123]. At the same time, the strong affinity of ion pairs is conducive to the rearrangement and association of physical crosslinking points [124,125]. Therefore, the introduction of ionic liquids or poly-ionic liquids into the preparation of self-healing polymers or composites has a good application prospect.

The ion–dipole interaction is another van der Waals force, which plays an important role in the self-healing process. Cao et al. [126] developed a self-repairing ionic conductor based on the ion–dipole interaction using Poly (vinylidene fluoride-co-hexafluoropropene) (PVDF-co-HFP) polymer and imidazolium salts (Figure 9a). The imidazolium salt possessed a high ionic strength and good electrochemical stability. The addition of imidazolium salt formed a highly reversible ion–dipole interaction, which promoted the cross-linking between the PVDF-co-HFP molecular chains. Furthermore, in the designed PVDF-co-HFP dynamic polymer networks, the addition of 1-ethyl-3-methylimidazolium trifluoromethanesulfonate (EMIOTf) could enhance the mobility of PVDF-co-HFP, which is conducive to the rearrangement of polymer chains on the fracture surface [127].

In the ECM systems, Huang et al. [128] developed a PVDF-co-HFP/EMIOTf/graphene ternary composite, which showed excellent interfacial bonding properties and self-healing performance (Figure 9b). Then, the ternary composite was utilized as a binder to prepare ECMs with 1-oxo-2,6-diamino-3,5-dinitropyrazine for further evaluation of the application potential in ECMs. The cracks generated during the Brazilian test could self-heal after 48 h at room temperature. The addition of graphene improved the thermal conductivity of ECMs.

**Figure 9 molecules-28-00428-f009:**
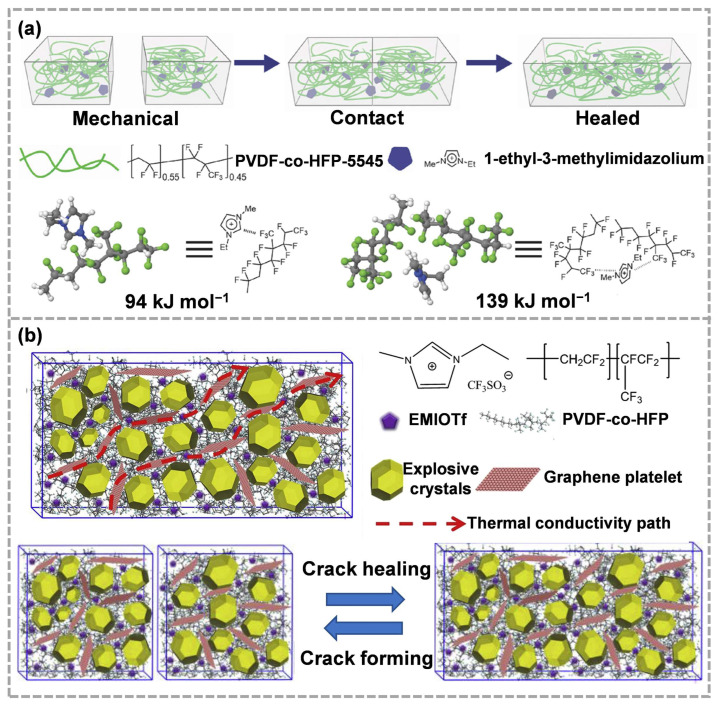
(**a**) Healing mechanism and chemical structures of polymers and imidazolium salts (reproduced from Ref. [126] with permission; Copyright Wiley, 2017), (**b**) the structure and self-healing of ECMs with PVDF-co-HFP/EMIOTf/graphene ternary composite as a binder (reproduced from Ref. [128] with permission; Copyright Elsevier, 2018).

The ion–dipole interaction opens a new direction for self-healing under mild conditions and even at room temperature. On the one hand, ionic liquids can have a plasticizing effect on polymer molecules and improve the mobility of molecular chains. On the other hand, due to the electrostatic interaction between anions and cations, there is no directivity and saturation, the repair efficiency is usually high and the repair conditions are mild. The combination of ion–dipole and other dynamic chemistry could further achieve reasonable regulation of mechanical properties and self-repair properties [129].

## 4. Challenges and Prospects

The research on self-healing binders for repairing microcracks/defects in ECMs has just started. At present, many well-designed strategies based on different dynamic chemistry have been born, such as dynamic covalent bonds based on Diels–Alder bonds, disulfide bonds and dynamic non-covalent bonds based on hydrogen bonds, metal–ligand coordination and ionic–dipole interactions.

Compared to the self-healing process caused by external stimuli (such as heating), room-temperature self-healing binders are more valuable because they simplify the repair process and reduce the cost. However, efficient self-healing properties are often related to the rapid diffusion of molecular chains and the rapid rearrangement of dynamic bonds [99]. Room-temperature self-healing polymers require more flexible molecular chains and weaker interactions, which is not conducive to mechanical properties [38]. For energetic self-healing binders, in addition to giving crack-healing abilities to ECMs, they also further improve the energy level. However, the existence of energetic groups as large-size side groups further weakens the mechanical strength [81]. Robust polymer networks can enhance the strength and toughness of binders but also hinder the diffusion of molecular chains and the rearrangement of dynamic bonds [107]. Therefore, how to realize the reasonable configuration of mechanical properties and self-healing efficiencies is a tough challenge.

In the future, in the field of ECMs, the development direction of self-healing binders may also include the aspects discussed in the following sections.

### 4.1. Introduction of Other Dynamic Chemical Methods

For most self-healing polymers, there is usually a trade-off between mechanical properties (tensile and toughness) and self-healing [110]. An ideal polymer network should have both robust microstructure and efficient dynamic exchange to achieve good mechanical properties and self-healing performance [130]. Therefore, more efforts are needed to focus on molecular engineering.

The coordination bonds with a large adjustable range of strength are conducive to adjusting the contradiction between mechanical properties and self-repair ability. Moreover, donor-acceptor self-assembly usually endows polymers with remarkable tensile, toughness and self-healing properties [131]. Interchain and intrachain donor-acceptor self-assembly can help to strengthen the microstructure of polymer materials similar to skeletal muscle proteins. Skeletal muscle proteins can absorb energy through reversible breaks of intramolecular secondary interactions, and then refold to induce recovery [132]. Similarly, donor-acceptor self-assembly can also be reassembled using external stimulation, providing tensile strength and self-healing capacity similar to human muscle [133]. For example, Ying et al. [48] introduced the electron donating structure of naphthalene ring (D motif) and the electron absorbing structure of imide group (A motif) into the main chain of polyurethane at the same time, thus synthesizing a polyurethane with donor and acceptor groups alternately distributed along the main chain (Figure 10a). The self-assembled structures between segment D and segment A endowed the obtained polyurethane with excellent toughness, anti-fatigue capability and remarkable room-temperature self-healing capabilities, which has important implications for self-repairing binders for ECMs.

### 4.2. Strategic Combination of Non-Covalent Bonds and/or Dynamic Covalent Bonds

For Diels–Alder bonds, ASCZ motifs and hindered urea bonds, the dynamic covalent crosslinking provide physical stability and strength, although the destruction of dynamic bonds needs to be carried out at high temperature. On the contrary, weak covalent or non-covalent bonds can be used to improve the dynamics of networks. Therefore, this strategic combination of strong and weak dynamic chemistry may solve the main conflict between mechanical properties and healing properties. For example, Guo et al. [134] developed a new self-healing material based on the Diels–Alder reaction (Figure 10b). Unique Diels–Alder adducts with appropriate dissociation and recombination dynamics were designed as cross-linking units to help facilitate processing and provide effective recovery time. The self-healing function was realized using a double dynamic network composed of disulfide bonds and hydrogen bonds. Hydrogen bonds and disulfide bonds belong to weak dynamic chemistry, which could self-heal at low temperatures while maintaining morphological integrity. In addition, hydrogen bonds and disulfide bonds also act as sacrificial bonds, dissipating energy and enhancing the tensile properties of materials. This provides a novel idea for designing self-healing binders for ECMs. Strong dynamic covalent bonds can be used as cross-linking units to improve the mechanical strength of the binders. At the same time, this dynamic chemistry provides a prerequisite guarantee for recycling and easy processing. In addition, the self-repair of ECM systems could be guaranteed using weak dynamic covalent bonds and non-covalent bonds.

### 4.3. Chemical Compatibility Evaluation of Self-Healing Binders with High Energetic Crystals

At present, in the application fields of ECMs, most reported self-healing binders lack the evaluation of compatibility with explosives. Incompatible reactions between explosives and polymers may lead to accelerated aging, reduced thermal stability, and accidental explosion due to decomposition reactions [135]. Therefore, chemical compatibility is an important index to evaluate the storage stability and reliability of ECMs, and it is also an important basis to evaluate whether there is a potential danger in the design, production and storage of ECMs. How to use reasonable thermal decomposition acceleration simulation experiments to accurately evaluate the stability and compatibility of samples with scientific and reasonable criteria has more practical significance. Currently, the methods for evaluating the compatibility of adhesives and explosives mainly include thermal analysis (including differential scanning calorimeter, differential thermal gravimetric analysis, thermostatic thermogravimetry and microcalorimeter analysis), gas analysis (including the Bourdon method and vacuum stability test) and mechanical sensitivity test, among which differential scanning calorimeter and the vacuum stability test are the most common [136,137,138].

## 5. Conclusions

In order to solve the problem that ECMs are prone to micro/fine cracks or defects on the bonding surface when exposed to external stimuli, substantial efforts have been devoted to conceiving construction strategies for self-healing binders. Internal self-healing is realized using dynamic chemistries, which possesses the intelligent attributes of self-sensing and self-adjusting. With these attributes, dynamic networks can autonomously respond to cracks/defects, and ultimately restore the functions of ECMs. Currently, binders based on dynamic covalent chemistries (Diels–Alder bonds, disulfide bonds) and non-covalent chemistries (hydrogen bonds, metal–ligand coordination, ion–dipole interactions) all have shown good self-healing abilities. Subsequently, in further application exploration, the self-healing binders played a positive role in the crack-healing of ECMs. In addition, the implantation of energetic groups into self-healing binders is conducive to further improving the energy level of ECMs.

However, mechanical properties and self-healing efficiencies are contradictory, and the large-size energetic side groups further weaken the mechanical strength. Therefore, the reasonable configuration of robust microstructures and effective dynamic exchange is still a challenge, which requires more effort in molecular engineering. The introduction of new self-healing functional units or the strategic combination of multiple dynamic chemistries are two directions to solve this challenge. In addition, the chemical compatibility between binders and explosives is recommended to be considered as an evaluation index for the stability of ECMs. These critical insights are emphasized to provide guidance for improving the stability, security and service life of ECMs.

## Figures and Tables

**Figure 1 molecules-28-00428-f001:**
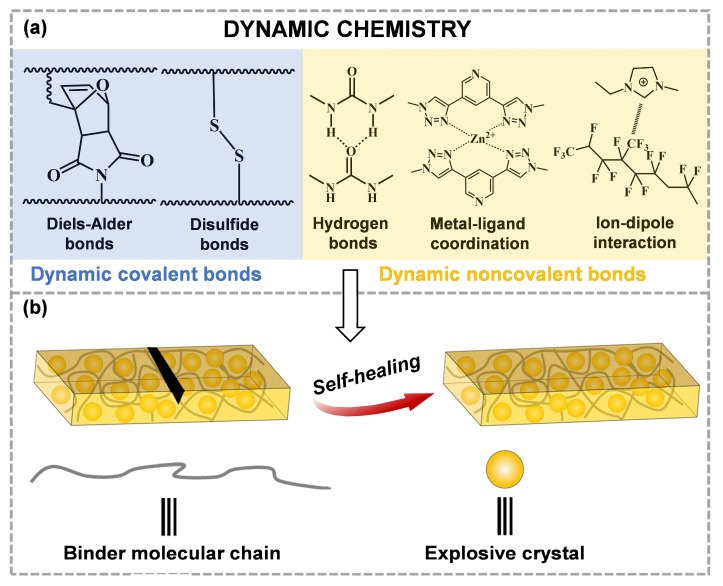
(**a**) Several typical self-healing methods for the crack-healing of ECMs, (**b**) the crack-healing process in ECMs.

**Figure 2 molecules-28-00428-f002:**
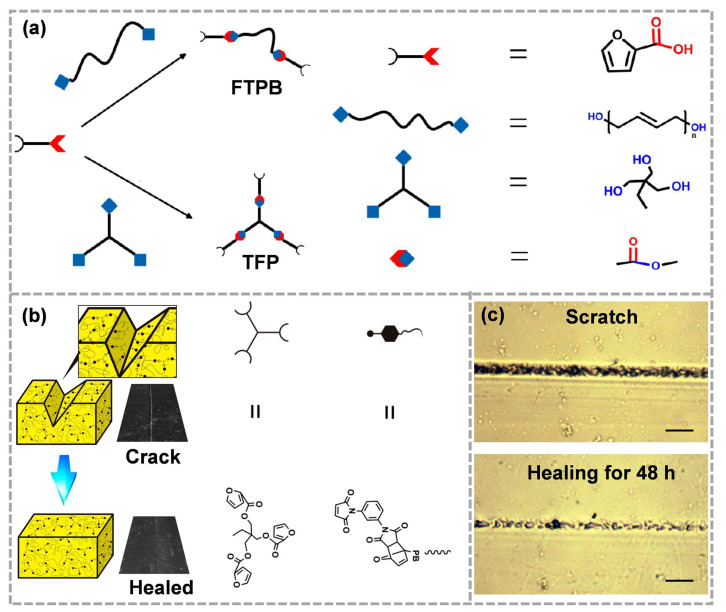
(**a**) Synthetic route of FTPB and TFP, (**b**) the self-healing process of TFP-FTPB-PDMI film and (**c**) the corresponding optical microscope images (reproduced from Ref. [57] with permission; Copyright MDPI, 2017).

**Figure 3 molecules-28-00428-f003:**
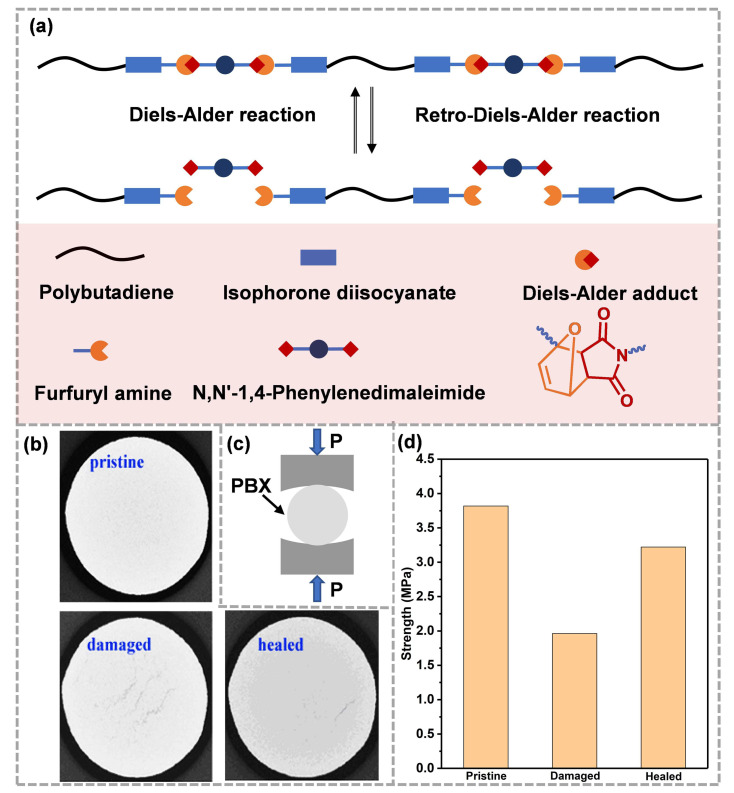
(**a**) Thermoreversible process of FTPB-DA for self-healing (reproduced from Ref. [58] with permission; Copyright Royal Society of Chemistry, 2021), (**b**) CT images, (**c**) Brazilian test method and (**d**) mechanical strength of DAPU-HMX (reproduced from Ref. [59] with permission; Copyright Elsevier, 2019).

**Figure 5 molecules-28-00428-f005:**
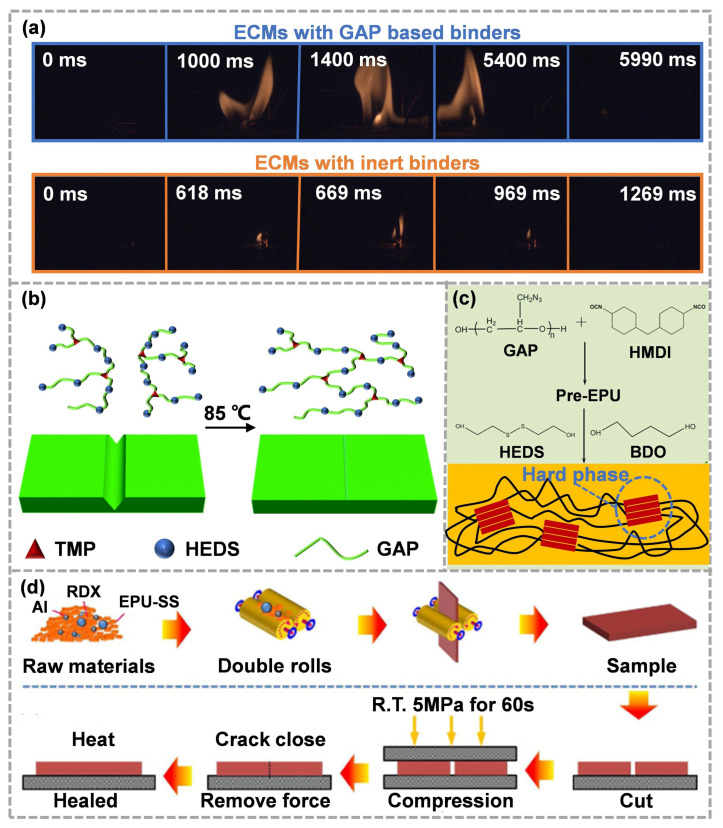
(**a**) Sequential open-combustion images of ECMs with GAP-based binders and ECMs with inert binders (reproduced from Ref. [81] with permission; Copyright Royal Society of Chemistry, 2021), (**b**) the self-healing process of GAPUV (reproduced from Ref. [82] with permission; Copyright Royal Society of Chemistry, 2021), (**c**) the synthesis process and microstructure of EPU-SS(reproduced from Ref. [83] with permission; Copyright American Chemical Society, 2021), (**d**) preparation and healing process of ECMs with EPU-SS as binders (reproduced from Ref. [84] with permission; Copyright Elsevier, 2022).

**Figure 8 molecules-28-00428-f008:**
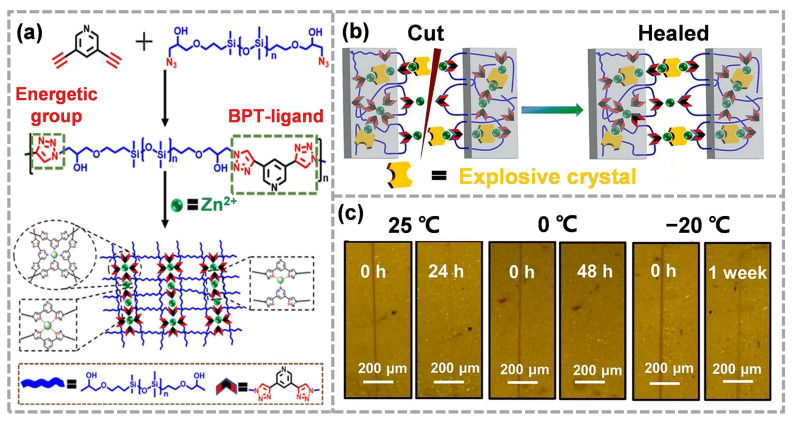
(**a**) Structural design and preparation of 3,5-BTP-PDMS-Zn polymer adhesive, (**b**) the self-healing mechanism for ECMs with 3,5-BTP-PDMS-Zn as binders, (**c**) optical microscope photographs of ECMs bonded with 3,5-BTP-PDMS-Zn demonstrating crack-healing (reproduced from Ref. [118] with permission; Copyright Elsevier, 2021).

**Figure 10 molecules-28-00428-f010:**
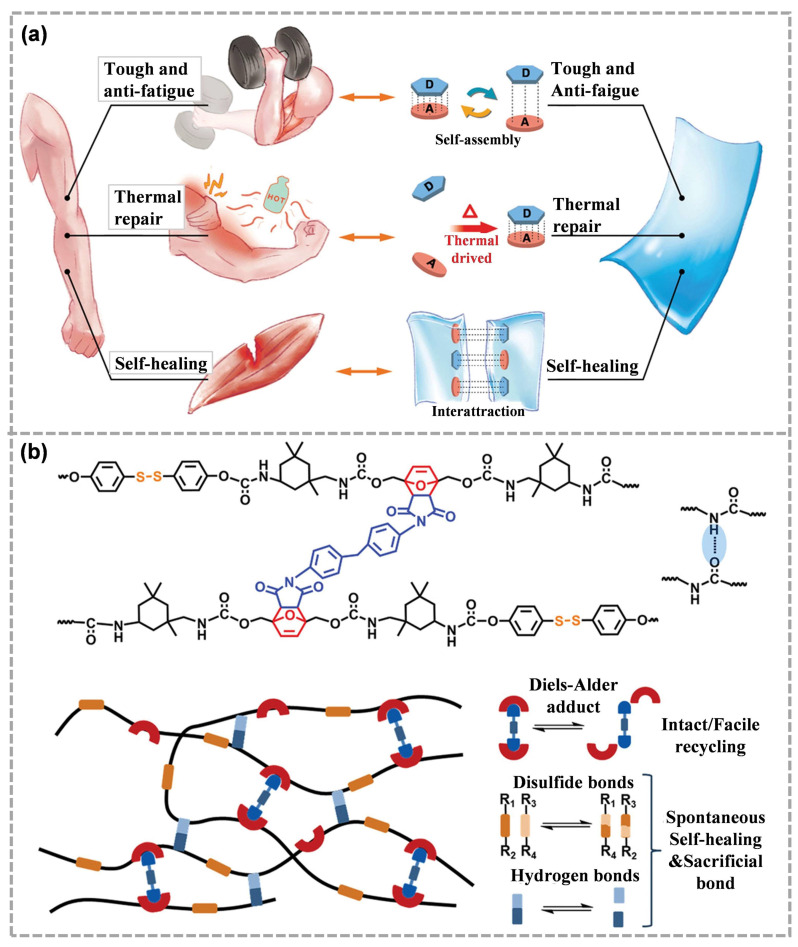
(**a**) Schematic diagram of muscle-inspired polyurethane-based donor-acceptor self-assembly (reproduced from Ref. [48] with permission; Copyright Wiley, 2021), (**b**) molecular structure and schematic structure of polyurethane including reversible covalent crosslinks based on Diels–Alder adducts, disulfide bonds and hydrogen bonds (reproduced from Ref. [134] with permission; Copyright Wiley, 2021).

## Data Availability

Not applicable.

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
