# Peer review of "Current Self-Healing Binders for Energetic Composite Material Applications"

_molecules, 2023, doi:10.3390/molecules28010428_

Round 1

Reviewer 1 Report

The paper presents a study on current self-healing binders for energetic composite material 2 applications

The following recommendations are proposed:

·         Please, revise the abstract to make it more accessible to a wider audience.

·         Please, improve the introduction section. It misses a comprehensive literature review.

·         Please, provide a flow chart of the paper organization.

·         Overall, English needs to be double-checked for typos.

·         The conclusions section can be better organized.

Main Concern:

What is the innovation that this paper brings into scientific knowledge?

Reviewer 2 Report

1) The abstract and conclusion sections need to be improved.

2) The novelty of the conducted research is not very strong.

3) The literature review is too general. Moreover, the literature gap is not discussed.

4) In present article, it needs to add the main research contributions of the present research compared with the existing research.

5) The deeper discussion of the obtained literature is necessary, in order to show the really contribution to this research area, when compared with the existent and studied literature. This is the key part of this review article.

6) It is found that the manuscript is incomplete in all sense and the main discussion section is minimum.

7) The article does not have a section that outlines/discusses the challenges in the proposed topic that should be expected from a proper review paper.

8) In conclusion, the paper does not provide some new analysis or findings in the proposed paper and hence much more efforts are necessary to present the existing knowledge in a way to benefit the readers and give them a proper idea about the current state-of-the-art in the field.

Reviewer 3 Report

Energetic composite materials are the basic materials of polymer adhesive explosive and composite solid propellant, as well as the symbol and key technology of modern weapon equipment upgrading. This review describes the research progress of self-healing binders in energetic composite materials systems; The structural design of these strategies to manipulate macro-molecular and/or supramolecular polymers will be discussed in detail, and then the im-plementation of these strategies on energetic composites will be discussed. Finally, the development direction of these self-healing material systems will prospect be prospected. The results look interesting and there is a good potential in the conducted research for industrial and fundamental research area. I believe there is a strong potential in this work and the manuscript structure and flow are good. However, some issues need to be addressed before the manuscript could be considered for publication. 1. Based on your literature review, it was not clear what the current question is for energetic composite materials. You very clearly understand what the authors you cited have done, and lack of in-depth mechanisms introduction; 2. I searched several another papers about ECM that published in AM, AFM, AEM, and NC. So I suggest the author do a more comprehensive literature search for this review; 3. Many Figures over the paper are not clear, and also unattractive, except for 1, the rest of which are dull, such as 2-10. 4. A well arrangement need to be conducted; 5. Define the scientific necessity of this review in abstract. 6. The conclusions in this manuscript are primitive. Please, write your conclusions.

Round 2

Reviewer 1 Report

Comments addressed

Reviewer 2 Report

The author considered some comments of reviewers. Nonetheless, several weaknesses were still found.

The manuscript entitles, “Current self-healing binders for energetic composite material applications”, shows high similarity index with already published data. As this is against the Copyright Act, consequently, it should not be accepted for publication in Molecules. It must be rewritten and submitted again for further evaluation.

It is necessary to reduce the level of similarity of the paper <10%.